# Parasite DNA and Markers of Decreased Immune Activation Associate Prospectively with Cardiac Functional Decline over 10 Years among *Trypanosoma cruzi* Seropositive Individuals in Brazil

**DOI:** 10.3390/ijms25010044

**Published:** 2023-12-19

**Authors:** Ashwin Sunderraj, Luisa Marin Cunha, Matheus Avila, Shaina Alexandria, Ariela Mota Ferreira, Léa Campos de Oliveira-da Silva, Antonio L. P. Ribeiro, Maria do Carmo Pereira Nunes, Ester C. Sabino, Alan Landay, Jorge Kalil, Christophe Chevillard, Edecio Cunha-Neto, Matthew J. Feinstein

**Affiliations:** 1Feinberg School of Medicine, Northwestern University, Chicago, IL 60611, USA; 2Faculdade de Ciências Médicas de Santos, UNILUS, Santos 11045-101, Brazil; 3Department of Preventive Medicine, Feinberg School of Medicine, Northwestern University, Chicago, IL 60611, USA; shaina.alexandria@northwestern.edu; 4Graduate Program in Health Sciences, State University of Montes Claros, Montes Claros 39401-089, Brazil; arielamota91@gmail.com; 5Institute of Tropical Medicine, University of São Paulo, São Paulo 05403-000, Brazil; lea.c.oliveira@gmail.com (L.C.d.O.-d.S.);; 6Department of Internal Medicine, Federal University of Minas Gerais, Belo Horizonte 31270-901, Brazil; tom1963br@yahoo.com.br (A.L.P.R.); mcarmo@waymail.com.br (M.d.C.P.N.); 7Division of Geriatrics and Gerontology, Department of Medicine, Rush University Medical Center, Chicago, IL 60612, USA; 8Laboratory of Immunology, Heart Institute Instituto do Coração (InCor), School of Medicine, University of São Paulo, São Paulo 05403-000, Brazil; jkalil@usp.br; 9Institut MarMaRa, TAGC Theories and Approaches of Genomic Complexity, Aix Marseille Université, 13385 Marseille, France; christophe.chevillard@univ-amu.fr; 10Division of Cardiology, Department of Medicine, Feinberg School of Medicine, Northwestern University, Chicago, IL 60611, USA

**Keywords:** Chagas disease, infection, immune response, heart failure, *Trypanosoma cruzi*

## Abstract

Parasitemia and inflammatory markers are cross-sectionally associated with chronic Chagas cardiomyopathy (CCC) among patients with *Trypanosoma cruzi*. However, the prospective association of the parasite load and host immune response-related characteristics with CCC (that is, progressors) among *T. cruzi* seropositive individuals has only been partially defined. In a cohort of *T. cruzi* seropositive patients in Montes Claros and São Paulo, Brazil who were followed over 10 years, we identified the association of a baseline *T. cruzi* parasite load and systemic markers of inflammation with a decline in cardiac function and/or the presence of cardiac congestion 10 years later. The progressors (*n* = 21) were individuals with a significant decline in the left ventricular ejection fraction and/or elevated markers of cardiac congestion after 10 years. The controls (*n* = 31) had normal markers of cardiac function and congestion at the baseline and at the follow-up. They were matched with the progressors on age, sex, and genetic ancestry. The progressors had higher mean parasite loads at the baseline than the controls (18.3 vs. 0.605 DNA parasite equivalents/20 mL, *p* < 0.05). Of the 384 inflammation-related proteins analyzed, 47 differed significantly at a false discovery rate- (FDR-) corrected *p* < 0.05 between the groups. There were 44 of these 47 proteins that were significantly higher in the controls compared to in the progressors, including the immune activation markers CCL21, CXCL12, and HCLS1 and several of the tumor necrosis factor superfamily of proteins. Among the individuals who were seropositive for *T. cruzi* at the baseline and who were followed over 10 years, those with incident CCC at the 10-year marker had a comparatively higher baseline of *T. cruzi* parasitemia and lower baseline markers of immune activation and chemotaxis. These findings generate the hypothesis that the early impairment of pathogen-killing immune responses predisposes individuals to CCC, which merits further study.

## 1. Introduction

Approximately 30% of *Trypanosoma cruzi*-infected individuals progress to chronic forms of Chagas disease, including chronic Chagas cardiomyopathy (CCC) [1], and a third of patients with CCC develop ventricular dysfunction, heart failure, arrhythmia, or sudden death. Cross-sectional investigations have revealed comparatively higher markers of inflammation and immune activation profiles among patients with CCC versus *T. cruzi* seropositive controls without CCC [2,3,4,5]. Moreover, detection of *T. cruzi* DNA in the blood has been associated with CCC, and the parasitic load has been found to inversely correlate with left ventricular dysfunction among CCC patients [6,7]. Broadly, these studies raise the hypothesis that a long-standing *T. cruzi* infection induces a pro-inflammatory milieu and may implicate Th1 cells and IFNgamma pathways in the pathogenesis of CCC; however, little is known regarding the prospective association of immune activation markers with cardiac decline in *T. cruzi* seropositive individuals. 

In this analysis, we evaluated circulating markers of immune activation and inflammation associated prospectively with worsening cardiac dysfunction and/or congestion after 10 years among *T. cruzi* seropositive patients. We hypothesized that the parasite DNA levels and the markers of immune activation and inflammation would be higher at the baseline for individuals who had progressive cardiac dysfunction at the 10-year marker compared with controls who remained free from cardiac dysfunction after 10 years.

## 2. Results

The baseline demographic and clinical characteristics were largely similar for progressors and controls, except for hypertension being more common in progressors (Table 1). The mean baseline *T. cruzi* DNA levels were significantly higher in progressors than in controls (18.3 vs. 0.605 DNA parasite equivalents/20 mL; *p* = 0.018). 

Of the 384 proteins analyzed, 47 had NPX levels that differed significantly across the two groups (Table 2; mean NPX of all proteins across both groups are listed in Appendix A). The levels were lower in the progressors compared to in the controls for 44 of the 47 proteins, including CCL21, CCL25, CD4, CXCL12, TNF, TNFSF12, and TNFSF13. The levels of the other three proteins (EIF5A, IDS, and JCHAIN) were higher in the progressors compared to in the controls (Figure 1).

In secondary analyses restricted to participants with an EF ≥ 55 at the baseline (to exclude those without potential pre-existing CCC with ventricular dysfunction), 39 of the 384 proteins had NPX levels that differed significantly across the two groups, including several that were lower among the progressors (CCL21, CXCL12, HCLS1, and TNFSF12). The levels of only one protein (JCHAIN) were higher in the progressors compared to in the controls in this analysis (Table 2).

Taken together, 31 proteins of the 384 proteins analyzed had NPX levels that differed significantly across both clinical groups in both the primary and secondary analyses (Table 2). Of these, 30 proteins showed higher expression in the controls than in the progressors, including CCL21, CXCL12, HCLS1, and TNFSF12. Only the levels of JCHAIN were elevated in the progressors vs. the controls in both analyses (Table 2).

## 3. Discussion

In a nested case–control study of a cohort of *T. cruzi* seropositive individuals followed prospectively for 10 years, we observed that those with progressive cardiac dysfunction and/or congestion had a higher baseline *T. cruzi* parasite load and lower baseline markers of immune activation and inflammation. 

Our findings ran counter to our central hypothesis that there would be higher indices of inflammation at the baseline among *T. cruzi* seropositive individuals who subsequently had progressive cardiac dysfunction, compared to those without progressive dysfunction. This hypothesis had been informed by cross-sectional studies that noted higher indices of inflammation among patients with prevalent CCC compared to controls without CCC [2,3,4,5]. Nevertheless, these prior studies were cross-sectional, leaving open the possibility that the heightened inflammation among patients who had already progressed to CCC relates to generally heightened inflammatory activation from heart failure [8]. Based on our observations of a comparatively higher parasite load and generally lower markers of immune response and inflammation among progressors versus controls, a plausible hypothesis generated from our findings is that the *T. cruzi* seropositive individuals with persistent parasitemia and reduced pathogen-killing immunity are at a higher prospective risk of cardiac decline. Prior investigations have shown that Chagas disease in general and CCC in particular are associated with reduced immune responsiveness to antigens during acute and chronic infections [9,10,11]. Moreover, they have shown that Chagas disease in general and CCC in particular are associated with dysfunctional/exhausted CD8+, CD4+, and CD4+CD8+ T cell responses, with increased membrane expression of inhibitory receptors, and with lower antigen-specific multifunctional capacity compared to that of asymptomatic patients [12,13]. Our group found that markers of potentially protective cytotoxic peripheral blood NK or CD8+ T cells were more highly expressed among moderate CCC patients without ventricular dysfunction, compared to severe CCC patients with ventricular dysfunction [14]. 

In addition, a *T. cruzi* infection is exacerbated in settings of impaired innate immune response [15,16,17]. Moreover, it has been found that chronic Chagas disease patients with positive blood *T. cruzi* DNA PCR displayed higher plasma levels of IL-10, a regulatory/immunosuppressive cytokine, than those with negative *T. cruzi* DNA PCR [18]. Our findings of an increased parasite load and decreased levels of several proteins associated with innate immune response and chemotaxis in the progressors vs. the controls, including CCL21, CXCL12, TNFSF12, and HCLS1, are potentially consistent with this body of literature. CCL21-CCR7 signaling is essential for IFN-gamma-mediated *Toxoplasma gondii* parasite clearance [19,20], and CCL21 mRNA is increased in end-stage CCC heart tissue [21]. CXCL12 also plays a role in protozoan parasite clearance [22,23] and shows increased levels in established CCC [24]. TNFSF12/TWEAK signaling activates NF-kB-mediated production of proinflammatory mediators [25], while HCLS1 is an established mediator more broadly of leukocyte recruitment and chemotaxis [26] and is an upregulated hub gene in hearts from established end-stage CCC patients and *T. cruzi*-infected mice [23]. Other markers that were comparatively less expressed at the baseline in the progressors included HCLS1, COLEC12, and MILR1, which is suggestive of impaired pathogen-induced leukocyte activation and antigen presentation [27,28,29,30]. Given our concomitant finding that the progressors displayed a higher amount of *T. cruzi* DNA in their peripheral blood than the controls, this may suggest that progressors are less capable of controlling the parasitism than the controls. The strengths of this study included a long-term follow-up of the participants for Chagas cardiomyopathy in a cohort with extensive clinical and sociodemographic profiling [31]. The key limitations included a limited sample size and a limited number of progressors, prompting the inclusion of individuals with some degree of LV dysfunction at the baseline as progressors, if their function became significantly more impaired at the follow-up. Additionally, some of the 31 proteins that differed in both analyses were minimally documented in the existing literature and did not have a clear putative relationship with the progression of Chagas disease. Notably, JCHAIN levels were found to be higher in the progressors than in the controls, and this protein has previously been noted to be expressed by mucosal and glandular plasma cells [32]. In addition, proteins were only measured at one point in time (the baseline) and not after 10 years at the follow-up. Moreover, we had a limited ability to account for any residual confounding in the regression analyses due to this sample size.

Nevertheless, our findings suggest that, although immune activation and inflammatory markers may be higher in individuals with prevalent CCC, this may not be the case prior to the development of CCC. Indeed, the results inform a new hypothesis that impairment in immune responses related to pathogen clearance early on—manifested here as decreased markers of immune response and inflammation—may predispose individuals to inferior parasite control and progression to CCC. Specifically, a markedly diminished inflammatory/immune response biomarker profile may develop early in the *T. cruzi* infection, prior to the onset of CCC/dilated cardiomyopathy, and a considerably more inflammatory profile may emerge once CCC/dilated cardiomyopathy develops. 

## 4. Materials and Methods

### 4.1. Study Design

We performed a nested case–control study of baseline-circulating protein biomarkers associated with decline in left ventricular function and/or cardiac congestion 10 years later. The participants in this study were recruited between 2008 and 2010 from cohorts of *T. cruzi* seropositive individuals at two sites in Brazil: São Paulo, São Paulo and Montes Claros, Minas Gerais [31]. The participants underwent standardized health questionnaires and medical evaluations at the baseline and at the 10-year follow-up, including their age, their sex, their hypertension and diabetes statuses, an electrocardiogram (ECG), an echocardiogram (Echo), and a phlebotomy with a cryopreservation of samples for subsequent blinded analyses of *T. cruzi* PCR and protein biomarkers. The blood samples were collected in EDTA and serum tubes and processed for parasite detection (described below) or spun and aliquoted. All specimens were frozen and maintained at −70 °C. 

The progressors were defined as individuals with a substantial decline in left ventricular function on the echocardiography between the baseline and 2018 and/or with evidence of cardiac congestion by 2018 as marked by elevated *N*-terminal pro-B-type natriuretic peptide (NT-proBNP > 500 pg/mL). This group of progressors is described in Appendix A (*n* = 21). The controls (*n* = 31) had normal echocardiographic, ECG, and NT-proBNP findings at the baseline and in 2018 and were propensity matched 2:3 with the progressors on age, sex, and ancestry, based on GWAS data [33]. One progressor and two controls from the original twenty-two and thirty-three, respectively, were excluded due to incomplete data. For the two groups, we compared the baseline *T. cruzi* parasitemia and the normalized protein expression levels of a panel of 384 proteins relevant to immune response and inflammation. 

### 4.2. Blood Bank Screening Procedures

The Fundação Pro-Sangue blood center performed *T. cruzi* antibody screening during the initial recruitment using three serological methods, including ELISA, hemagglutination, and immunofluorescence, described previously [34,35]. The participants were included if they were positive in all three assays at the time of the donation. In Montes Claros, the Hemominas blood center screened blood donations with ELISA and hemagglutination. Donors who were positive in both assays at the time of the donation were considered to be eligible for this study.

### 4.3. Parasite DNA Detection

At the time of the baseline interviews and examinations, 20 mL of EDTA-anti-coagulated blood was collected from each enrolled subject and immediately mixed with an equal volume of 6 M of guanidine HCl–0.2 M EDTA solution, boiled for 15 min, and ultimately vortexed and aliquoted into (1.0 mL) aliquots for the qPCR reactions, as described previously [8]. 

### 4.4. Serum Biomarkers

We analyzed the normalized protein expression (NPX) levels from a multiplex immunoassay of 384 inflammatory biomarkers (384 analytes, plus internal controls per panel, Olink Explore 384, Olink Bioscience, Uppsala, Sweden) [36]. The NPX levels were expressed on a log2 scale whereby a one unit increase in the NPX level indicated a doubling of the protein concentration. Each NPX variable was centered to have a mean of 0 and scaled to have a standard deviation of 1 before analysis.

### 4.5. Statistical Analysis

We compared protein expression in the progressors versus the controls using two-sided *t*-tests to identify the proteins with significantly different expressions across the groups. To account for multiple comparisons, a false discovery rate- (FDR-) adjusted *p*-value of <0.05 was required for the protein expression levels to be considered significantly different between the groups. The primary analyses were performed in the complete case group (21 progressors and 31 controls). To investigate whether the associations remained when considering only individuals with a normal LVEF at the baseline, we repeated this comparison in the subgroup of patients with an LVEF > 55% at the baseline (*n* = 43, *n* = 12 progressors, and *n* = 31 controls). The differences in the mean NPX levels and the corresponding FDR-adjusted *p*-values for each analysis are reported below.

## Figures and Tables

**Figure 1 ijms-25-00044-f001:**
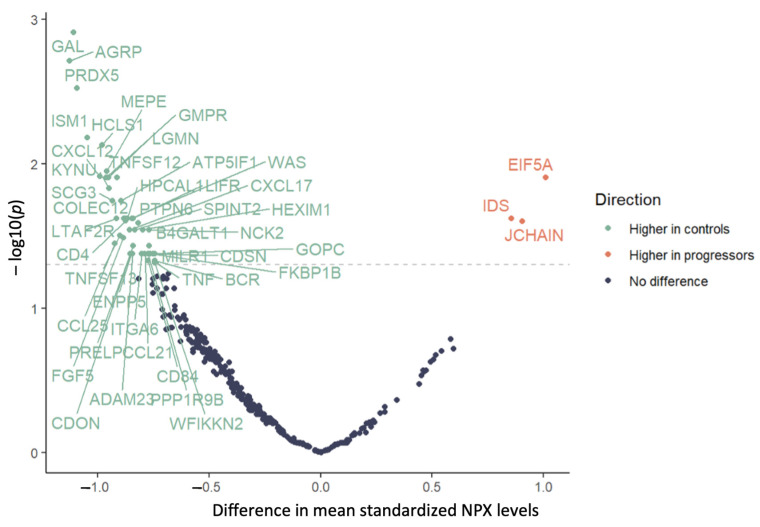
Mean difference in normalized protein expression levels in progressors vs. controls; dashed line indicates *p* = 0.05, and statistically significant mean differences indicated above the line.

**Table 1 ijms-25-00044-t001:** Baseline Characteristics.

Characteristics	CONTROLS (*n* = 31)	PROGRESSORS (*n* = 21)	*p* Value
Sex, *n* (%)			1
Male	18 (58.1%)	13 (61.9%)	
Female	13 (41.9%)	8 (38.1%)	
Age at T0 ^1^, Mean (SD)	42.8 (7.65)	48.2 (9.75)	0.04
Age at T1 ^2^, Mean (SD)	51.6 (7.60)	56.3 (9.75)	0.07
Hypertension at Baseline			<0.01
No	31 (100%)	11 (52.4%)	
Yes	0 (0%)	10 (47.6%)	
Hypertension at Follow-up			0.03
No	30 (96.8%)	16 (76.2%)	
Yes	1 (3.2%)	5 (23.8%)	
Diabetes at Baseline			0.68
No	28 (90.3%)	18 (85.7%)	
Yes	3 (9.7%)	3 (14.3%)	
Diabetes at Follow-up			1
No	27 (87.1%)	18 (85.7%)	
Yes	4 (6.5%)	3 (14.3%)	
EF ^3^ at Baseline, Mean (SD)	62.5 (2.59)	53.3 (8.43)	<0.01
EF ^3^ at Follow-up, Mean (SD)	67.6 (4.90)	39.2 (12.8)	<0.01
*T. cruzi* DNA ^4^ at T0, Mean (SD)	0.605 (1.63)	18.3 (43.6)	0.02

^1^ Time 0, including baseline medical assessment and interview. ^2^ Time 1, at 10-year follow-up. ^3^ Ejection fraction (%). ^4^
*T. cruzi* DNA measured in DNA parasite equivalents/20 mL.

**Table 2 ijms-25-00044-t002:** Proteins with significantly different normalized expression levels for progressors versus controls (referent).

	Primary Analysis *n* = 52			Restricted Analysis ^1^ *n* = 43		
Protein	Change	Mean Difference	*p* Value ^2^	Change	Mean Difference	*p* Value ^2^
ADAM23	Decrease	−0.842	0.042	–	−0.758	0.084
AGRP	Decrease	−1.125	0.002	Decrease	−1.073	0.012
AMN	–	−0.71	0.102	Decrease	−1.054	0.040
ATP5IF1	Decrease	−0.894	0.018	Decrease	−1.073	0.001
B4GALT1	Decrease	−0.768	0.037	Decrease	−0.923	0.012
BCR	Decrease	−0.743	0.047	Decrease	−0.973	0.001
CCL21	Decrease	−0.788	0.042	Decrease	−1.017	0.010
CCL25	Decrease	−0.924	0.036	–	−0.915	0.097
CD4	Decrease	−0.874	0.025	–	−0.815	0.073
CD79B	–	−0.705	0.060	Decrease	−0.702	0.028
CD84	Decrease	−0.773	0.047	–	−0.863	0.052
CDON	Decrease	−0.84	0.037	–	−0.845	0.067
CDSN	Decrease	−0.743	0.042	–	−0.659	0.111
COLEC12	Decrease	−0.874	0.024	Decrease	−0.908	0.026
CXCL12	Decrease	−0.99	0.012	Decrease	−1.094	0.026
CXCL17	Decrease	−0.856	0.029	–	−0.667	0.149
EIF5A	Increase	1.01	0.012	–	0.798	0.145
ENPP5	Decrease	−0.848	0.042	–	−0.847	0.077
F2R	Decrease	−0.883	0.024	Decrease	−1.022	0.006
FGF5	Decrease	−0.884	0.032	–	−0.916	0.077
FKBP1B	Decrease	−0.752	0.042	Decrease	−0.897	0.013
FOXO1	–	−0.603	0.134	Decrease	−0.818	0.020
GAL	Decrease	−1.108	0.001	Decrease	−1.17	0.001
GMPR	Decrease	−0.962	0.012	Decrease	−0.993	0.012
GOPC	Decrease	−0.765	0.042	Decrease	−0.905	0.006
HCLS1	Decrease	−0.979	0.007	Decrease	−1.017	0.006
HEXIM1	Decrease	−0.798	0.029	Decrease	−0.9	0.012
HPCAL1	Decrease	−0.867	0.024	Decrease	−0.907	0.012
IDS	Increase	0.856	0.024	–	0.649	0.149
ISM1	Decrease	−1.045	0.007	Decrease	−1.08	0.006
ITGA6	Decrease	−0.802	0.042	–	−0.816	0.069
JCHAIN	Increase	0.904	0.025	Increase	1.124	0.014
KYNU	Decrease	−0.949	0.015	Decrease	−1.032	0.012
LGMN	Decrease	−0.952	0.012	Decrease	−0.977	0.026
LIFR	Decrease	−0.841	0.024	Decrease	−0.938	0.024
LTA	Decrease	−0.915	0.024	–	−0.83	0.054
MEPE	Decrease	−0.959	0.011	Decrease	−0.964	0.010
MGMT	–	−0.633	0.121	Decrease	−0.797	0.043
MILR1	Decrease	−0.744	0.042	Decrease	−0.807	0.026
NBN	–	−0.491	0.206	Decrease	−0.672	0.042
NCK2	Decrease	−0.77	0.029	Decrease	−0.912	0.006
PPP1R9B	Decrease	−0.773	0.042	Decrease	−0.868	0.015
PRDX5	Decrease	−1.093	0.003	Decrease	−1.055	0.012
PRELP	Decrease	−0.854	0.042	–	−0.708	0.077
PTPN6	Decrease	−0.818	0.026	Decrease	−0.971	0.011
SCG3	Decrease	−0.933	0.018	Decrease	−0.974	0.015
SCRN1	–	−0.734	0.060	Decrease	−0.841	0.049
SMOC2	–	−0.411	0.282	Decrease	−0.762	0.040
SPINT2	Decrease	−0.832	0.029	Decrease	−0.93	0.012
TNF	Decrease	−0.738	0.049	–	−0.635	0.102
TNFRSF4	–	−0.349	0.395	Decrease	−0.736	0.040
TNFSF12	Decrease	−0.914	0.012	Decrease	−0.914	0.026
TNFSF13	Decrease	−0.901	0.031	–	−0.837	0.077
WAS	Decrease	−0.846	0.024	Decrease	−0.897	0.010
WFIKKN2	Decrease	−0.744	0.049	Decrease	−0.983	0.011

Proteins with decreased expression in progressors vs. controls in both primary and restricted analyses are shaded blue, while those with increased expression in progressors vs. controls in both primary and restricted analyses are shaded red. ^1^ Restricted to participants with an ejection fraction ≥ 55% at baseline. ^2^ FDR adjusted.

## Data Availability

The data presented in this study are openly available in https://biolincc.nhlbi.nih.gov/studies/chagas/ (accessed on 23 November 2023).

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
