# Peer review of "Parasite DNA and Markers of Decreased Immune Activation Associate Prospectively with Cardiac Functional Decline over 10 Years among Trypanosoma cruzi Seropositive Individuals in Brazil"

_ijms, 2023, doi:10.3390/ijms25010044_

Round 1

Reviewer 1 Report

Comments and Suggestions for Authors

In this article, a 10-year follow-up study was conducted on cardiac progressive patients seropositive for Trypanosoma cruzi, in comparison to a control group. The assessment of biomarkers in peripheral blood reveals certain increases in molecules among the patients when compared to the controls. However, what is most striking is the reduction in inflammatory markers among patients who progress. Given the limited number of studies demonstrating inflammatory markers in Chagas disease and their association with cardiac disease progression, certain aspects of the article require further substantiation to suggest that a reduction in systemic inflammation is linked to the progression of cardiac disease.

The introduction is indeed quite brief and lacks definition of certain key aspects of the study. It could be valuable to briefly mention some of the theories concerning the progression of cardiopathy and highlight relevant prior studies.

Regarding Table 1, it is advisable to provide descriptions for some of the characters used in the footnotes, even if they may appear repetitive. Additionally, a statistical analysis when comparing the displayed variables would be beneficial.

Both references concerning immunosuppression in Chagas disease are outdated for the research conducted in recent years. Furthermore, one of them is based on animal models.

Similarly, the references explaining the low function of chemokines in cardiac progressive patients are primarily drawn from the context of other diseases such as malaria or toxoplasmosis.

The study involving the measurement of markers in the serum of Chagas disease patients appears to contradict numerous studies conducted in humans, where some of the TNF-alpha inflammatory molecules produced in response to the parasite have been associated with disease progression and cardiac damage. How do the authors reconcile these differences?

The authors must provide more robust support for their findings in this manuscript in order for it to be considered for publication.

Comments on the Quality of English Language

Minor editing of English language required

Reviewer 2 Report

Comments and Suggestions for Authors

Major comments

1) Section 4.4 & Figure 1. There is a cut-off line at approximately 1.3 on the y-axis of Figure 1. The authors should explain in the text how this cut-off value was calculated.

2) Line 128. Is JCHAIN one of the minimally documented proteins? The authors should if possible give some more information on JCHAIN in the Discussion, as they have mentioned this protein twice in the Results section.

3) Table 1. The authors shoud add units to the EF values, and explain this abbreviation (Ejection Fraction).

4) Table 1. The authors should say ‘T. cruzi DNA’ not ‘Chagas DNA’, and also state the units (parasite equivalents/20 ml) as in Line 68.

Other comments

Line 4/28/30/31/32/42/43/45/50/54/56/60/67/94/96/99/120/138/150/162/165. Trypanosoma cruzi or T. cruzi must be in italic font.

Line 28/50. Delete ‘(T. cruzi)’.

Line 36. Is there a missing word, for example: ‘…and were MATCHED with progressors…’?

Line 47. Add Trypanosoma cruzi to the Keywords.

Line 124. The authors state that sociodemographic profiling was done, but no socio- status data is given here. If they are referring to their published work, it should be cited here.

Line 150. Is ‘pro-‘ missing before ‘B-type’? (compare Line 156).

Table 1. In the Control and Progressor groups, it appears that the numbers of cases of diabetes identified at baseline had decreased at follow-up. If this is correct, the authors should mention this in the text.

Table 1. The authors should consider if the ‘Total’ column is useful, as they have carefully segregated the study group into Controls and Progressors.

Table 2. The column title ‘Restricted analysis’ has superscript 1, which is currently not explained.

Table 2. The authors should clarify that proteins that were ‘decreased’ in the Primary and Restricted analyses are indicated in grey colour rows here.

Suppl Table 1. The column title ‘Protein’ has superscript 1, which is currently not explained.

Reviewer 3 Report

Comments and Suggestions for Authors

Title: Parasite DNA and markers of decreased immune activation associate prospectively with cardiac functional decline over 10 years among Trypanosoma cruzi seropositive individuals in Brazil.

In the present manuscript, Sunderraj A et al. performed a nested study comparing selected proteins and DNA parasite levels in individuals from Brazil with Chagas disease, specifically with and without chronic Chagas cardiomyopathy. The results suggested higher levels of proteins related to the immune response in those patients without a progression to the chronic cardiac form of the disease, which coincided with significantly lower levels of parasitic DNA in these patients.

This is a study with interesting results in which the effort of sample collection and follow-up for 10 years is noteworthy. Nevertheless, there are some aspects that need to be improved and clarified as they would help the understanding of the manuscript.

MAJOR COMMENTS

1.- While at the time of collection T.cruzi DNA levels were measured in both progressors and controls, I was wondering if there are recorded data on the parasite DNA levels of these patients over the course of the 10 years and, if so, if significant changes were observed both within and between patient groups.

2- Similar to the previous comment, I was wondering if a comparison of the levels and expression of the different proteins was performed longitudinally in each group of patients, i.e., were there significant changes of these values in the progressors group over time, and same within the controls?

3.- Supplementary Figure 1 indicates that some individuals were excluded from further analyses due to lack of covariate data. Were other variables such as age, sex, or clinical data included as covariates in the statistical analyses? Related to this, Table 1 indicates the presence or absence of Diabetes in the patients, what is the reason for including this cofactor and not others? This should be clarified in the Materials and Methods section.

4.- In the methodology and sample description it is not clear if the sample information, clinical data and the parasite DNA levels were recruited for all patients at T0 and 10 years later, i.e., if all this information was collected in both time points. This should be clarified in the Materials and Methods section.

MINOR COMMENTS

1.- The scientific name of the parasite should be written in italics.

2.- In Figure 1, the x-axis should indicate that the protein levels are expressed on a log2 scale (similar to how the y-axis has been indicated).

3.- EF in Table 1 should be defined.

4.- It is assumed that the brown row in Table 2 corresponds to the protein that showed an increased expression levels; however, the pattern of the rest of the colors is not clear. The color code of the rows in this table should be explained in the table footer.

5.- Regarding Table 2, there are exponents 1 and 2 in "Restricted Analysis" and "P value" but it is not indicated what they refer to. This should be clarified in the table footnote.

Round 2

Reviewer 2 Report

Comments and Suggestions for Authors

I thank the authors for the clarifications in the revision, and there are no further comments from me. Congratulations on this interesting work.

Author Response

We thank the reviewer for their kind comments!

Reviewer 3 Report

Comments and Suggestions for Authors

While some of the recommendations have been addressed by the authors, there are some minor changes that should be addressed in the manuscript:

- It is not clear if this is due to the track changes, but in several parts of the manuscript there are typos in the references insertion, so that they are after the spelling marks. For example, line 61: seropositive controls without CCC [2-5]. Moreover, detection of T. cruzi DNA in blood ...

- There are still some parts of the text where the names of the parasites (i.e., T. cruzi and Toxoplasma gondii) are not in italics.

- The footnotes in Table 1 are misattributed. Numbers 2 and 3 are not identified.

Author Response

Reviewer 3

- It is not clear if this is due to the track changes, but in several parts of the manuscript there are typos in the references insertion, so that they are after the spelling marks. For example, line 61: seropositive controls without CCC [2-5]. Moreover, detection of T. cruzi DNA in blood ...

We thank the reviewer for this attention to detail. We have reformatted our citations in accordance with IJMS guidelines.

- There are still some parts of the text where the names of the parasites (i.e., T. cruzi and Toxoplasma gondii) are not in italics.

We have searched the manuscript for references to parasite names, and corrected them to italics as needed.

- The footnotes in Table 1 are misattributed. Numbers 2 and 3 are not identified.

We thank the reviewer for this comment, and have corrected this error in Table 1.